# Multi-Level Model to Predict Antibody Response to Influenza Vaccine Using Gene Expression Interaction Network Feature Selection

**DOI:** 10.3390/microorganisms7030079

**Published:** 2019-03-14

**Authors:** Saeid Parvandeh, Greg A. Poland, Richard B. Kennedy, Brett A. McKinney

**Affiliations:** 1Tandy School of Computer Science, University of Tulsa, Tulsa, OK 74104, USA; parvandehsaied@gmail.com; 2Mayo Vaccine Group, Mayo Clinic, Rochester, MN 55905, USA; poland.gregory@mayo.edu (G.A.P.); kennedy.rick@mayo.edu (R.B.K.); 3Department of Mathematics, University of Tulsa, Tulsa, OK 74104, USA

**Keywords:** gene interaction, vaccine immune response, nested cross-validation

## Abstract

Vaccination is an effective prevention of influenza infection. However, certain individuals develop a lower antibody response after vaccination, which may lead to susceptibility to subsequent infection. An important challenge in human health is to find baseline gene signatures to help identify individuals who are at higher risk for infection despite influenza vaccination. We developed a multi-level machine learning strategy to build a predictive model of vaccine response using pre−vaccination antibody titers and network interactions between pre−vaccination gene expression levels. The first-level baseline−antibody model explains a significant amount of variation in post-vaccination response, especially for subjects with large pre−existing antibody titers. In the second level, we clustered individuals based on pre−vaccination antibody titers to focus gene−based modeling on individuals with lower baseline HAI where additional response variation may be predicted by baseline gene expression levels. In the third level, we used a gene−association interaction network (GAIN) feature selection algorithm to find the best pairs of genes that interact to influence antibody response within each baseline titer cluster. We used ratios of the top interacting genes as predictors to stabilize machine learning model generalizability. We trained and tested the multi-level approach on data with young and older individuals immunized against influenza vaccine in multiple cohorts. Our results indicate that the GAIN feature selection approach improves model generalizability and identifies genes enriched for immunologically relevant pathways, including B Cell Receptor signaling and antigen processing. Using a multi-level approach, starting with a baseline HAI model and stratifying on baseline HAI, allows for more targeted gene−based modeling. We provide an interactive tool that may be extended to other vaccine studies.

## 1. Introduction

One of the major challenges of machine learning models that use omic data to predict influenza vaccine immune response is defining the outcome. Ideally, individuals in training data would be labeled as protected and unprotected following vaccination. Instead, hemagglutination inhibition (HAI) titer is often used as a surrogate marker of protection (e.g., HAI titer of 1:40). In vaccinomic research, an increase or fold-change (day 28/day 0 HAI) is often used to assess vaccine efficacy (e.g., high and low responders based on 4-fold change relative to pre−vaccination titer) [1]. However, HAI fold change by itself may be misleading and modeling should account for pre−existing HAI for influenza [2]. For example, subjects with high pre−vaccination influenza HAI titers show little increase in titer following vaccination but appear to be protected [3]. In other words, when there is a previous exposure, a small fold change may simply mean that ceiling effects or feedback from the immune system limit the amount the immune system can increase pre−existing antibody titers. Thus, instead of using fold change as a binary response in statistical models of immune response, we choose to model the variation in post-vaccination HAI as a function of previous HAI levels in addition to omic predictors.

Before using genomic information, the initial stage of our multi-level approach used an inverse model of baseline HAI to predict post-vaccination HAI. Previous studies have addressed the inverse correlation of initial titers on day-28 fold-change observed in influenza vaccination studies [4,5,6] using the adjusted max fold change (AdjMFC) [7]. AdjMFC and MFC are defined in greater detail in Reference [7]. To summarize, one first computes the maximum titer for individual virus strains that are standardized (subtracting the median followed by dividing the maximum absolute deviation (MAD) across subjects within each virus strain). To compute adjMFC, one bins subjects based on their maximum baseline titers (among virus strains) and then subtracts the median and divides by the MAD within each bin. The nonlinear correlation between MFC and day 0 titers is removed by first dividing the cohort into groups of subjects with similar day 0 titers and then adjusting the response within each group so that groups are normalized and, thus, comparable. Finally, for each metric, “high” and “low” responders are defined as subjects ranked above or below the top or bottom 20th percentile mark of the adjMFC, respectively. Other studies [8] have used 30% and 70% thresholds to create three groups of low, medium, and higher antibody responses.

Our approach, rather than discretize the response, attempts to use all variation in day-28 HAI response. After modeling with baseline HAI levels, we discretized the baseline (pre−vaccine) HAI values and stratified subjects into low, medium, and high prior-exposure groups because we expect different gene regulatory mechanisms to be involved in each group. For the high prior-exposure group, we did not train a gene−based model because most of the variation in the HAI response is already explained by a simple baseline HAI model.

The adjMFC used in Reference [7] to classify subjects into high and low responders, excludes medium responders which may lead to loss of power due to exclusion of subjects. In addition, power may be lost due to discretization of response data. Thus, we used a regression strategy because dichotomization of the outcome variable (e.g., classifying subjects as responders or non-responders based on 4-fold change in HAI from day 0 to day 28) can lead to loss of statistical power [9,10]. In the first stage of analysis, we used a simple model of day-28 response from day-0 HAI and then used the residuals of this model to train a day-0 gene expression model to explain additional variation in day-28 HAI.

The primary aim of the current study is to use baseline transcriptomic profiles to improve prediction of vaccine immune response. In a recent comparison of microarray-based classifiers, the quality of prediction depended on the phenotypic outcome more than the particular machine learning approach [11]; hence, our attention to the definition of HAI outcome is based on a residual day-0 HAI model. The inverse day-0 Ab titer model that we used in the initial stage of analysis explains a significant amount of variation in day-28 response, especially for vaccinees with large day-0 antibody titers. However, for those with low day-0 Ab titers, there is a large amount of unexplained variation in day-28 titers. As immune responses are regulated by transcriptomic activity in immune cell subsets, this variation is likely controlled by gene interactions.

To capture gene interaction effects, we used a feature selection algorithm called regression-based genetic association interaction network (reGAIN) feature selection that finds pairwise interaction or differential co-expression effects [12]. In previous work, we identified replicating modules for depression based on co-expression [13]. Here we used interaction effects as opposed to correlation to find important predictors. We used the ratios of the top gene pairs as predictors in the predictive HAI model to create more generalizable models as the relative changes in expression are more consistent between cohorts. To reduce the risk of overfitting with reGAIN feature selection we used nested cross-validation [14,15,16] and an independent test set to report titer predictions. We trained and tested the multi-level approach on multiple publicly available influenza vaccine gene expression studies. Our results suggest that the reGAIN feature selection approach improves model generalizability and identifies genes enriched for immunologically relevant pathways. Using a multi-level approach, starting with a baseline HAI model and stratifying based on baseline HAI, allows for a piecewise model that flexibly targets genes involved in different pathways based on prior exposure. We provide an interactive tool that may be extended to other vaccine studies.

## 2. Materials and Methods

### 2.1. Overall Approach

Our analytical pipeline involved four main modeling steps (Figure 1). To provide a broad overview, these modeling steps begin with (A) a nonlinear model of post-vaccination fold-change as a function of the day-0 HAI levels using an inverse power model to predict the post-vaccination HAI. This initial model captures a large amount of variation in post-vaccination response, especially those with high day-0 HAI. We next (B) identified subjects with low, medium, and high day-0 HAI for additional stratified gene−based modeling. Subjects were clustered using their day-0 HAI titers by Gaussian mixture modeling (GMM). The day 0 HAI titers in all data sets were combined and the GMM generates three clusters of observations: low, medium, and high baseline levels (red, green, blue in Figure 1B). A first level of gene expression feature selection was performed using the reGAIN method (C) for the low (red) and medium (green) baseline titer groups. Genes were not used to model the high baseline group (blue) because little additional variation is explained beyond the day-0 HAI model. The reGAIN method was used in nested cross-validation (CV) to select 200 genes with the highest interaction scores from the pairwise gene−gene interaction regression coefficients. A second level of feature selection (D) with cross-validated glmnet, uses the residuals of the nonlinear model (from A) as a new outcome variable, and genes and gene ratios from (C) were used as predictor variables. The Baylor data was used as training data, and the Emory and Mayo data were used for testing.

#### 2.1.1. Inverse Power of Baseline HAI to Model Post-Vaccine HAI Fold Change

One of the major challenges to identifying genes and signatures that influence vaccine immune response is defining the immune response outcome and accounting for pre−existing immunity (day-0 HAI). Thus, before predicting immune response from omic data, we determined the explained immune response variation based on easily observable variables, such as day-0 HAI and age. It is known that the post-vaccination change in response is negatively correlated with pre−vaccination titers, and regression approaches, such as the titer response index (TRI), have been used to adjust for this effect [4]. Feng et al. noted that the change in post-influenza vaccination antibody is inversely related to pre−existing antibody in multiple groups, including elderly and subjects with lupus [17]. We used an inverse power law model of post-vaccine HAI fold-change (Figure 1A), $fc\left(d_{o}\right)=a\;d_{o}{}^{b}$, where fc is the predicted fold change (day28 HAI/day0 HAI) and d_0_ is day0 (pre−vaccine) HAI. We fit the model parameters *a* and *b*, where *b* is expected to be negative.

#### 2.1.2. Expectation Maximization/Gaussian Mixture Model

We used Gaussian mixture model (GMM) density estimation [18] to cluster subjects based on pre−vaccination HAI. The GMM algorithm estimates a finite mixture of models using maximum likelihood estimation and expectation maximization methods. For these clusters, we created piecewise regressions models that predict HAI fold change based on gene expression for each baseline group separately (Figure 1B). This stratified model building allows for the selection of genes most relevant to modeling vaccine response within each prior exposure group. We bypassed gene−based modeling for the high baseline group because little additional variation is explained beyond the day-0 HAI model in the first stage. 

#### 2.1.3. reGAIN Gene−Gene Interaction Based Feature Selection of Baseline Gene Expression

A regression-based genetic association interaction network (reGAIN) is a statistical network that encodes the pairwise statistical interactions between genes A and B conditioned on an outcome variable Y [19,20,21].
(1)$$Y\sim \;\beta_{0}+\beta_{1}A+\beta_{2}B+\beta_{3}AB+\sum_{i=1}^{n}\beta_{3+i}COV_{i}+\varepsilon .$$

Each element of the weighted network is constructed from the interaction coefficient β_3_ between each pair of genes. The full model corrects for the main effects of variables A and B (β_1_ and β_2_) and for possible covariates (COV_1_—COV_n_) that may be relevant to the outcome. For dichotomous phenotypes, the interaction term represents a differential co-expression between genes and in this case reGAIN uses logistic regression (Figure 1C). In the current study, the outcome Y in the second-level modeling stage is the residual of the HAI fold change from the day-0 HAI model (Figure 1A). Thus, the outcome is continuous, and linear regression is used. The most statistically significant interactions were selected for subsequent machine learning (Figure 1D). In addition, we wrapped reGAIN feature selection into nested cross-validation to prevent overfitting as discussed next.

#### 2.1.4. Baseline Gene Expression Machine Learning Model to Improve Baseline HAI Model

Bootstrapping and cross-validation (CV) provide reliable internal validation and estimation of classification accuracy [9,22]. Because the gene−gene interaction features from reGAIN incorporate phenotype information, we used a nested CV procedure to avoid biased classification errors and to properly evaluate the generalized performance of classifiers [14]. Because reGAIN feature selection computes interactions or differential correlation effects, we used the ratio of genes as features in the machine learning classifier (Figure 1D). Using the relative expression level between two genes within a data set may improve generalizability to other data sets by reducing the effect of systematic differences between samples.

We used glmnet hyperparameter tuning with CV [14] for the final model of HAI fold change. glmnet tunes the α [0, 1] and λ hyperparameters by CV, where a lower value of α leads to ridge penalty and a higher value leads to lasso penalty. A low value for λ generally includes many features and leads to overfitting. In addition to internal validation, we validated the machine learning model’s predictive ability on an independent data set. Given the relatively large size of the Baylor data (e.g., *n* = 200+ subjects; Table 1), an alternative to cross-validation is to split the data into three parts: a feature selection set, a training set, and a testing set. A 3-way split is also conducive to a differential privacy approach that uses threshold-out in a training and holdout data sets [23,24]. We provided R code and a Shiny app to reproduce this pipeline (https://github.com/insilico/predictHAI) and (http://insilico.utulsa.edu/predictHAI).

## 3. Results

### 3.1. Gene Expression and HAI Training and Testing Data

We trained and tested the proposed methods using three public datasets (Table 1) to build models of vaccine response using the multistage modeling strategy (Figure 1). These studies include virus-neutralizing titers H1N1 A/California/07/2009, A/Brisbane/59/07, H3N2 A/Uruguay/716/07, A/Perth/16/2009, B/Brisbane/60/2001, and B/Brisbane/3/2007. Reported titers were the highest dilution that completely suppressed virus replication. Not all data is available at each time point for all studies. For example, the Emory 2007–2009 data (GSE29619) consists of 63 subjects age 22 to 40 years old and includes baseline or pre–vaccination gene expression data but not the entire longitudinal gene expression data [25]. They showed that, even without vaccine−perturbed expression levels, it is possible to achieve good immune response prediction from baseline data [7,26]. Similarly, we used baseline gene expression with reGAIN machine learning feature construction. Another Emory study 2009–2011 (GSE74817) consists of 89 subjects age 21–85 years old vaccinated with TIV and available HAI in days 0, 1, 3, 7, 14, and only baseline gene expression [26]. We also used data from the gene expression omnibus (GEO) data from Baylor (GSE48018 and GSE48023) [4]. The Baylor data has a relatively large number of samples: approximately 100 healthy adult males and 100 healthy adult females with expression time series (Day 0, 1, 3, 14) and HAI (Day 0, 14, 28). The Mayo RNA-Seq gene expression data study consists of 105 old individuals from 57 to 92 years old (both genders) performed at the Mayo Clinic (Rochester, MN—available on ImmPort under study number SDY67). Note that the Baylor data set is a relatively young cohort (ages 19 to 41 years), whereas the Mayo and Emory data also include older subjects with related immunosenescence, which could affect prediction. Furthermore, the Mayo gene expression is derived from RNA-Seq while the other data are derived from microarray. We used quantile normalization for each dataset [27].

### 3.2. Day-0 HAI Model of Post-Vaccination HAI Fold Change

We trained the inverse power law (desensitization) model (Figure 1A and Section 2.1.1) of influenza HAI fold-change response from day-0 HAI using the Baylor data set (Table 1). We also tested a term that is linearly decreasing with age, but we did not include it in the final model because the adjustment has a very minor effect on the predictions. We tested the model on the Emory and Mayo data (Table 1). We applied a log_2_ transformation to all titers. Following the approach used by Tan et al. [1] to create the outcome variable, we used the maximum fold change (day28/day0) HAI antibody response across the three influenza vaccine strains (A-H1N1, B, and A-H3N2) for each subject. For the day-0 predictor in the Baylor training data, we used the day-0 levels from the strain that shows the max fold change for each subject (matched strains). The Mayo test data only includes A-H1N1 titers, so there is only one choice for the fold change and day 0 HAI values. The model on the Baylor, Emory, and Mayo data (Figure 2) explains 45%, 32%, and 22% of the variation in the maximum titer fold change when we trained day-0 HAI of the Baylor data. Thus, the model describes the general behavior of the fold-change as a function of day-0 HAI, but, for a given day-0 HAI, additional variation in the fold-change may be explained by gene expression data. Due to the older age of subjects in the Mayo data and the concomitant higher day-0 HAI, the Mayo fold-change is low compared to the other data sets (Figure 2).

### 3.3. GMM for Day-0 Clustering

We hypothesize that different baseline genes will be involved in predicting vaccine response depending on the individual’s baseline HAI (Figure 3). The inverse day-0 HAI model works best for subjects with high pre−existing titer, while more variation remains unexplained for subjects with medium or lower day-0 HAI. The post-vaccine response of subjects with low previous response shows much greater variability in fold change due to additional immune system mechanisms and factors. These low previous-response individuals will be of most interest in identifying additional variation through gene expression contributions. Thus, before gene expression modeling, we first defined three groups of day-0 HAI subjects. Using GMM clustering (Figure 1B, overall approach), we identified three normal densities of day-0 HAI observations (Figure 4). We set the number of clusters or mixtures to three (low, medium, high). We determine the cutoff points by the maximum posterior probability that each observation belongs to one of the Gaussian densities. To evaluate the robustness of the clusters, we combined day-0 HAI from all data sets (Table 1).

### 3.4. reGAIN for Constructing the Interaction Network

We used a data-driven, vaccine−specific network method called reGAIN [21] that incorporates interactions between genes that affect outcome variation [19,20,28]. We used the β_3_ interaction term in Equation (1) to construct the weighted network. Such supervised feature selection procedures carry the risk of overfitting; however, generalized performance is achieved by performing feature selection and training within each k-fold of cross-validation (CV). Embedded feature selection methods, such as glmnet, perform feature selection during training, and feature selection methods were also carried out within each cross-validation fold, but they are separate from the training. We used nested CV with 10 inner loops and 10 outer loops to compute reGAIN gene scores. In each fold, we selected the top genes with highest interaction values from the reGAIN matrix and filtered the top 200 genes with the highest frequency to create a gene−network based gene set for immune response prediction (Appendix A). In addition, we used the STRING network to characterize gene interactions (Appendix A).

We used pathway enrichment to characterize the biological function of the genes selected by reGAIN from baseline expression in the Baylor discovery data. We used the Molecular Signatures Database (MSigDB) [29] to compute the overlap of genes in known pathways with a query list of genes to assess a common biological function of the query gene set. We identified enriched Reactome pathways (Table 2) from a query list of the top 200 genes from nested CV. The p-values were calculated based on the hypergeometric distribution, which estimates the probability of observing a gene overlap ratio by chance compared to the ratio of the size of the pathway to the total number of genes. Many of the pathways related to the reGAIN network have clear relevance to predicting vaccine immune response: adaptive immune system, MHC-mediated antigen presentation, and B cell receptor signaling.

We trained the overall model (day-0 HAI plus day-0 gene interactions) on Baylor data and tested on Emory and Mayo (Figure 5). Additional variation is explained by gene interactions relative to the model with only day-0 HAI (Figure 2). The gene expression model shows the most improvement for the medium day-0 HAI group and overestimates the fold change for the low day-0 HAI group. Gene expression levels were not used for the high day-0 HAI group. We compared the observed versus predicted fold-change HAI in each group, low, medium, and high responders in Appendix A.

To compare the effect of using reGAIN feature selection, we carried out the same pipeline with a non-reGAIN feature selection method on the same data. For comparison, we used coefficient of variation (CoV) filtering, which is the ratio of the standard deviation to the mean of a gene. A gene with low CoV may be a useful predictor because its average magnitude is large and/or its effect is consistent across samples. It does not use the outcome variable for filtering, so it does not increase the risk of overfitting. Using the same data and modeling strategy as the reGAIN approach (Figure 5) we compared the results of modeling based on CoV filtering (Figure 6). We found the reGAIN feature selection model through nested CV leads to more variation explained (higher *R*^2^) in the independent data sets, suggesting that reGAIN helps find more biologically relevant combinations of genes.

### 3.5. Shiny Application

We developed a flexible and user-friendly tool for the methods developed as a Shiny application in the R statistical language. There are two different ways to use the Shiny app: (i) web-based, using a web application that is hosted on our web server and can be found at http://insilico.utulsa.edu/predictHAI/ or (ii) installed and run locally from https://github.com/insilico/predictHAI. The application is designed in eight stages (Figure 7). In each stage, the sidebar is designed for settings and pipeline customization, and a tabbed panel is designed to run the stages (run button) and display results. In stage-1 (Figure 7A) users are able to select the train and test data in the sidebar and “Run day-0 HAI modeling” to execute the day-0 model and display the plots. As described in the methods (Figure 1A), stage-1 uses an inverse power law model to train baseline HAI. We kept the remaining explained variation of each subject as a fold-change residual to use as the outcome variable for baseline gene expression modeling. Stage-4 was an intermediate stage in the pipeline that uses Gaussian mixture modeling to define the clusters (red, green, and blue densities) of subjects based on the baseline HAI values (Figure 7B). The boundary of low, medium, and high baseline HAI subjects was based on the maximum posterior probability using the expectation maximization method, where it can be seen by two vertical dashed lines.

Stage-6 executed the baseline gene expression modeling separately for the lower baseline−HAI responders and medium baseline−HAI responders (Figure 7C). The left column illustrates observed fold-change residuals (x-axis) and predicted fold-change residuals (y-axis), and the right column illustrates the variation explained using baseline gene expression prediction (red crosses) on fold-changes (black circle). Since we have two prediction model—the baseline HAI model and the baseline gene expression model—we created a table of *R*^2^ results using day-0 HAI modeling and day-0 gene expression modeling in stage-8 (Figure 7C). In stage-8, there are supplemental links in the sidebar that allow users to download the top 200 gene list, source code and data. The boxplots and all other plots used in the current study are created in the intermediate stages of the application. The application reacts to user changes up to the current stage, including changing the train and test data sets or parameters, such as the number of filtered genes or glmnet penalty. Currently the tool does not allow uploading of data due to resource allocation; however, this capability can be turned on by developers in the available source code.

## 4. Discussion

In this study, we introduced a multi-level modeling strategy for predicting immune response to influenza vaccine from baseline HAI and gene expression. To account for pre−existing exposures, our first-level model to predict post-vaccination HAI was an inverse power function of day-0 HAI. This model was trained on the Baylor data and explains post-vaccine fold change variation best for individuals with high pre−existing HAI titers. However, for individuals with low and medium pre−existing HAI, a model for the unexplained variation is needed that incorporates additional biological information. Thus, we developed a second-level day-0 gene expression model to explain additional variation in post-vaccine HAI fold change. We used the fold-change residuals from the first-level baseline HAI model (Figure 1) as the regression outcome for the second-level gene expression model.

Tsang et al. [7] standardized their day-0 serological and B-cell variables by the z-score transformation between day 0 and late post-vaccination to ensure that the different variables can be combined in different clustering group analysis. They used hierarchical and k-means clustering to cluster the subjects, and they assigned the subjects to three groups of high, medium, and low. They then excluded the medium group and consider the high and low groups for their further analysis. The group [8] used 30% and 70% percentile confidence intervals to determine thresholds for creating low, medium, and higher antibody responder groups. Excluding the medium group has the advantage of focusing on the extremes of response. On the other hand, additional response variation may be explained by baseline HAI titers if more groups are included. In our first-stage HAI-based model, we used all HAI baseline groups: high, medium, and low. In our second-stage gene−based model, we excluded the high day-0 baseline subjects from gene expression modeling because their first-stage model, with day-0 HAI titer alone, was already a good predictor of response.

For the remaining day-0 responders (low and medium), we filtered to 5000 genes across all data sets with coefficient of variation. We again used the Baylor data for training because it had the largest sample size. We used the regression-based genetic association interaction network (reGAIN) method with nested cross-validation (CV) for feature selection in the Baylor data. We selected the top 200 genes that have the highest interaction values based on the reGAIN matrix in each inner-loop of the nested CV in the training of glmnet. We chose the top 200 genes with highest frequency across nested CV folds because this leads to more stable features for modeling. The reGAIN method is able to detect functional interactions, which are likely to be important in immune system regulation, but other feature selection approaches and other machine learning algorithms have their own advantages and may be used in conjunction with this multi-level approach.

We used the ratio of gene pairs from reGAIN as predictors to help create more generalizable models because the relative changes in expression may be more consistent between samples and cohorts than changes of single genes. Using all possible ratios of these genes as predictor variables, we trained a glmnet model to predict fold-change residuals from the first-level model. We used 10-fold cross-validation to optimize the α [0, 1] and λ hyperparameters. We ran the gene−model pipeline for lower and medium baseline subjects separately. We did not run for higher baseline subjects because the baseline model is effective at predicting response to the vaccine. We tested our multi-level model on two independent data sets. The results show that our strategy with reGAIN feature selection with day-0 gene expression can explain additional variation in HAI fold-change response to vaccination compared with a non-reGAIN filtering.

Relevant pathways were enriched in the top 200 genes (Table 2 and Table 3), including Class I MHC Mediated Antigen Processing Presentation and Signaling by B cell receptor (BCR). In a previous study, a BCR signaling module was found to have a significant association with influenza vaccine immune response in a multi-cohort study [8]. Two of the gene sets linked to HAI response involved the processing and presentation of antigen. We speculate that a subject’s ability to process and present antigen may impact humoral immune responses to influenza vaccination by enhancing T cell help. It has been suggested that increased activation of T follicular helper cells may contribute to the development of higher titer Ab in response to the high dose influenza vaccine [30]. The presence of the BCR signaling pathways suggests that robust activation of antigen-specific B cells is also critically important for the developing Ab response. For example, B cells receiving maximal BCR stimulation may undergo more robust clonal expansion, have a higher propensity to develop into memory cells, and may produce greater quantities of antibody. It has been shown that influenza vaccination of older subjects results in the development of fewer vaccine−specific plasmablasts compared to younger recipients. This decrease is accompanied by lower concentrations of Abs but is not accompanied by any change in Ab avidity or affinity [31].

Our results suggest that one method to overcome this immunosenescence may be to provide optimal BCR signaling. It has also been demonstrated that the high dose influenza vaccine elicits a more robust plasmablast response than the standard dose vaccine [32]. The results from our study may provide mechanistic insights into the biological processes controlling the magnitude of the immune response to influenza vaccination.

We provide source code and a web-based tool to perform these analyses. The web-based tool is meant to interactively demonstrate the approach but currently does not allow upload of additional data. The web-based tool may be improved by allowing upload and additional preprocessing options on a suitable server. These capabilities are present in the provided source code. Additional variation may be explained by our model by incorporating prior knowledge from functional networks [16] and by integrating other types of biomarkers such as proteomics, genetic variants and methylation. However, a model that uses only baseline gene expression would be clinically more practical. Additional variation may be explained by other types of feature selection and by boosting the gene expression machine learning on individuals that lie farther away from the baseline HAI model prediction.

## Figures and Tables

**Figure 1 microorganisms-07-00079-f001:**
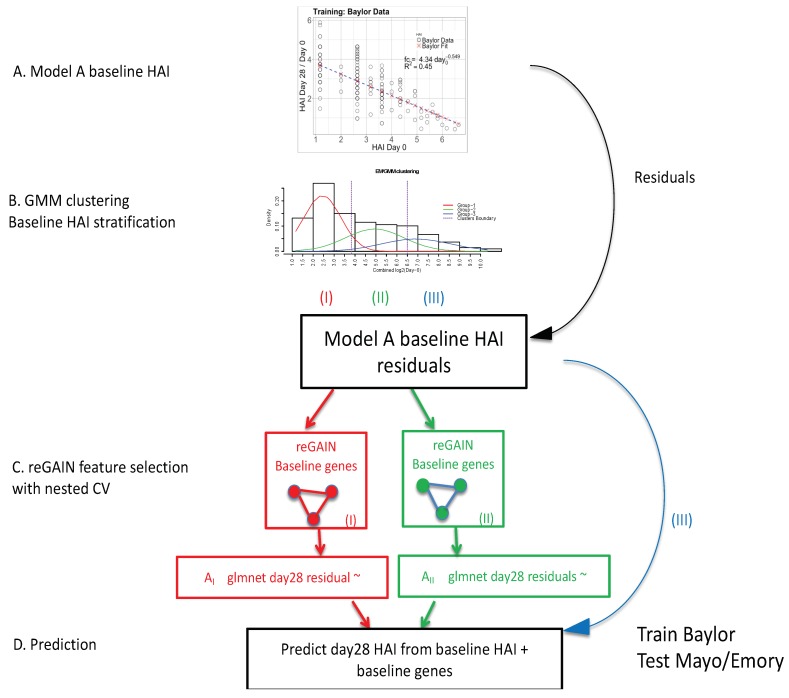
A diagram of the overall modeling approach in four parts. (**A**) Inverse power model is used to model the pre−existing hemagglutination inhibition (HAI) antibody titer (day-0 HAI). The horizontal axis is the day-0 HAI, and the vertical axis is the ratio of day-28/day-0 (fold-change) according to the inverse power model. (**B**) Day-0 HAI in all data sets are combined and clustered into three groups (low (red I), medium (green II), and high (blue III)) using Gaussian mixture modeling (GMM) clustering. (**C**) The regression-based genetic association interaction network (reGAIN) method is used to compute the weighted matrix for gene pairs through 10-outer/10-inner nested cross-validation (CV), and top 200 interacting genes with highest reGAIN values are selected as predictor variables in two of the baseline groups (low/red and medium/green). We excluded the third group (high day-0, blue) since the power law model works well in that group. (**D**) Final gene modeling using glmnet penalized regression, where the dependent variable is the residual from A, and predictor variables are the top 200 gene ratios from C. In gene modeling, we again trained on the Baylor data and tested on Emory and Mayo data for each group, separately.

**Figure 2 microorganisms-07-00079-f002:**
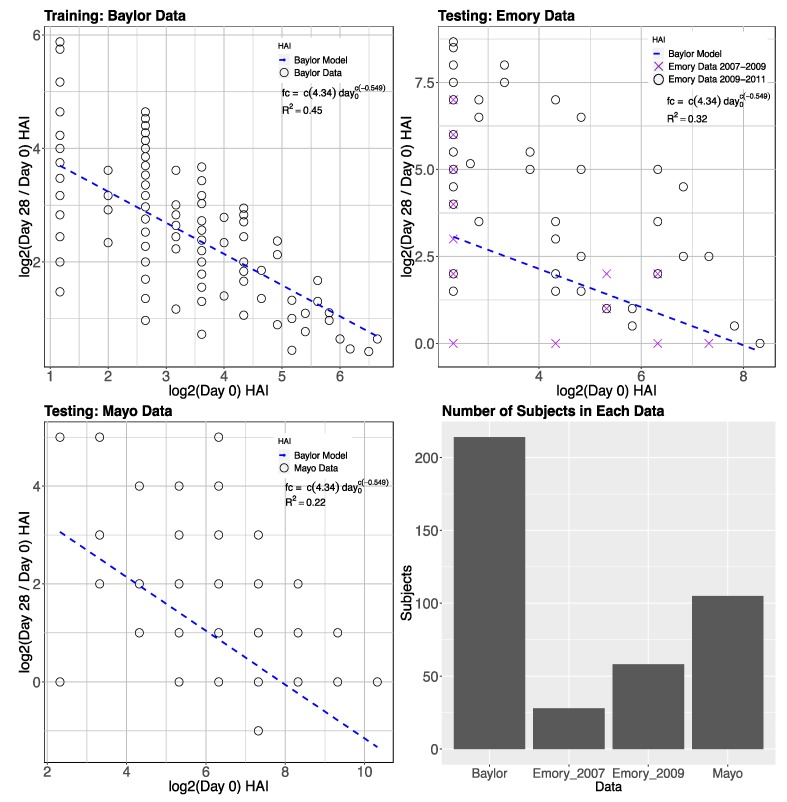
Baseline (day-0) HAI model prediction of HAI fold change (day 28) trained on the Baylor data and tested on Emory and Mayo data. The plots are on log_2_ scale, where predicted values are plotted with blue dash line with parameters trained from the Baylor data. The parameters are shown in the model equation along with the *R*^2^. The variation explained in fold change HAI is 45% in the Baylor data, 32% in the Emory data, and 22% in Mayo data. Emory data are a combination of two data sets from different years that are distinguished by black circles (2007–2009) and purple crosses (2009–2011).

**Figure 3 microorganisms-07-00079-f003:**
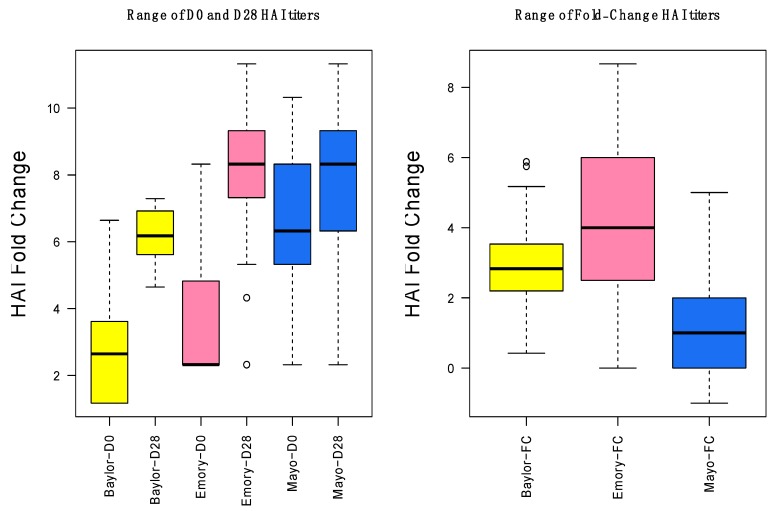
Range of day-0, day-28, and fold change HAI titers for the three data sets. The vertical axis shows the range of the HAI titers on a log_2_ scale for all data sets. Fold changes for Mayo are lower because the population was older and had higher initial HAI.

**Figure 4 microorganisms-07-00079-f004:**
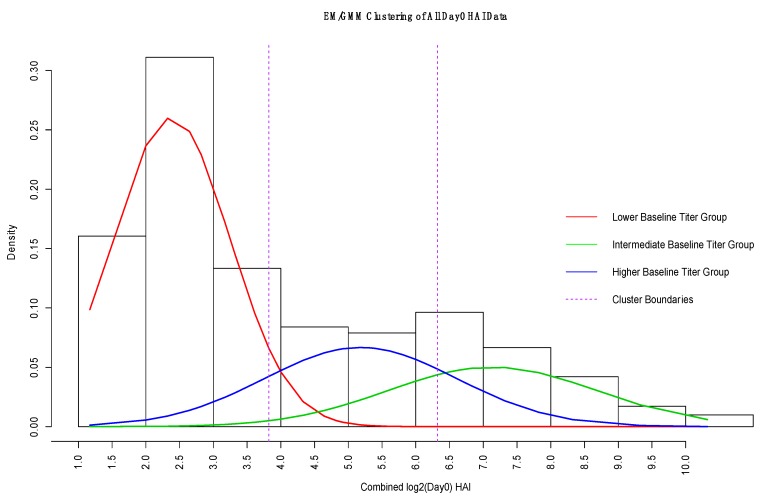
Gaussian Mixture Model (GMM) estimate of densities of day-0 HAI (log_2_) for combined data. The red density represents the lower baseline titer group, the green density is the intermediate baseline group, and the blue density represents the higher baseline titer group. The vertical dashed lines indicate the different group boundaries based on the maximum posterior probability. We used these boundaries to identify the cutoff points for creating the low, medium, and high day-0 titer groups for gene−based modeling.

**Figure 5 microorganisms-07-00079-f005:**
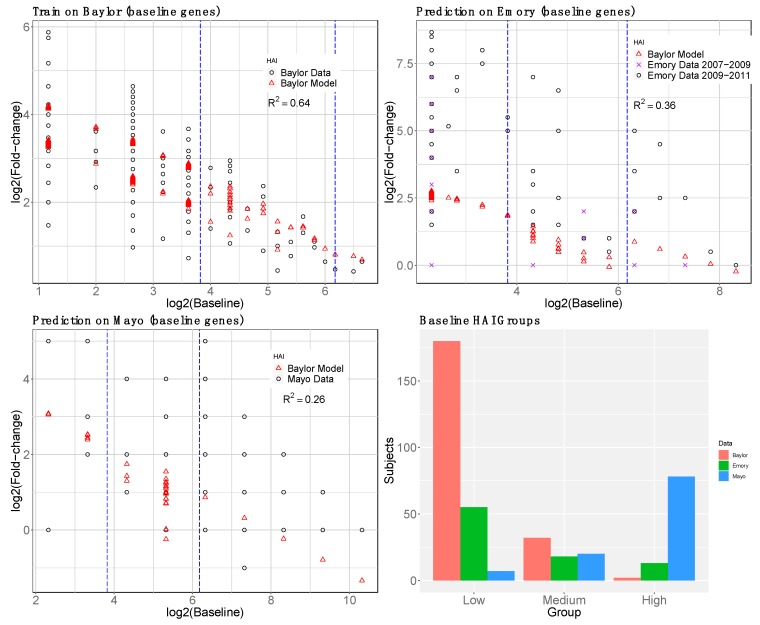
Second-stage (HAI plus gene expression) baseline (day-0) model prediction of post-vaccination HAI fold change (day 28). Genes were selected by reGAIN from baseline in nested cross-validation to predict the residual HAI fold change from the pre−vaccination HAI model (Figure 2). The model was trained on Baylor data (upper left). The black circles show the original data, and the red-triangles show the predicted outcome. The Emory prediction (upper right) consists of two data sets distinguished by black circles and purple x’s. The validation *R*^2^ is shown for each prediction. The barplot shows the number of subjects in each day-0 HAI titer group.

**Figure 6 microorganisms-07-00079-f006:**
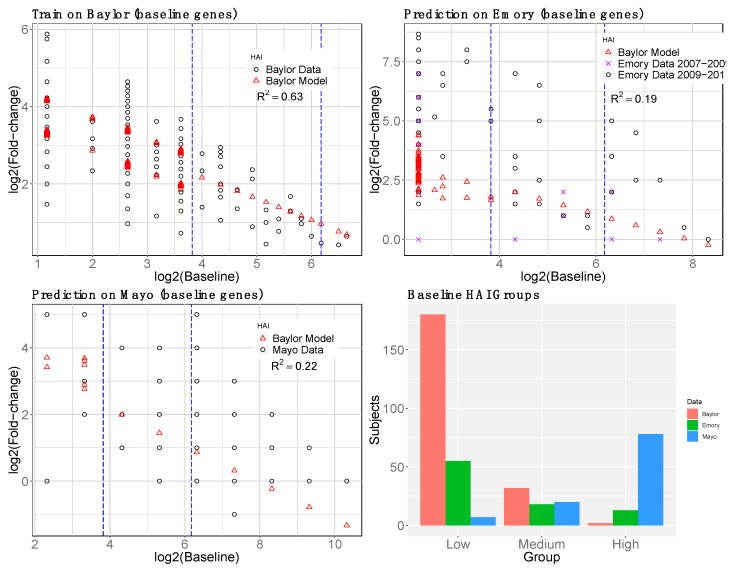
Non-reGAIN filtering used in the second-stage (gene) model prediction of post-vaccine HAI. Same as Figure 5 but instead of reGAIN feature selection, coefficient of variation was used to filter genes. The *R*^2^ for non-reGAIN is lower than reGAIN for training and testing.

**Figure 7 microorganisms-07-00079-f007:**
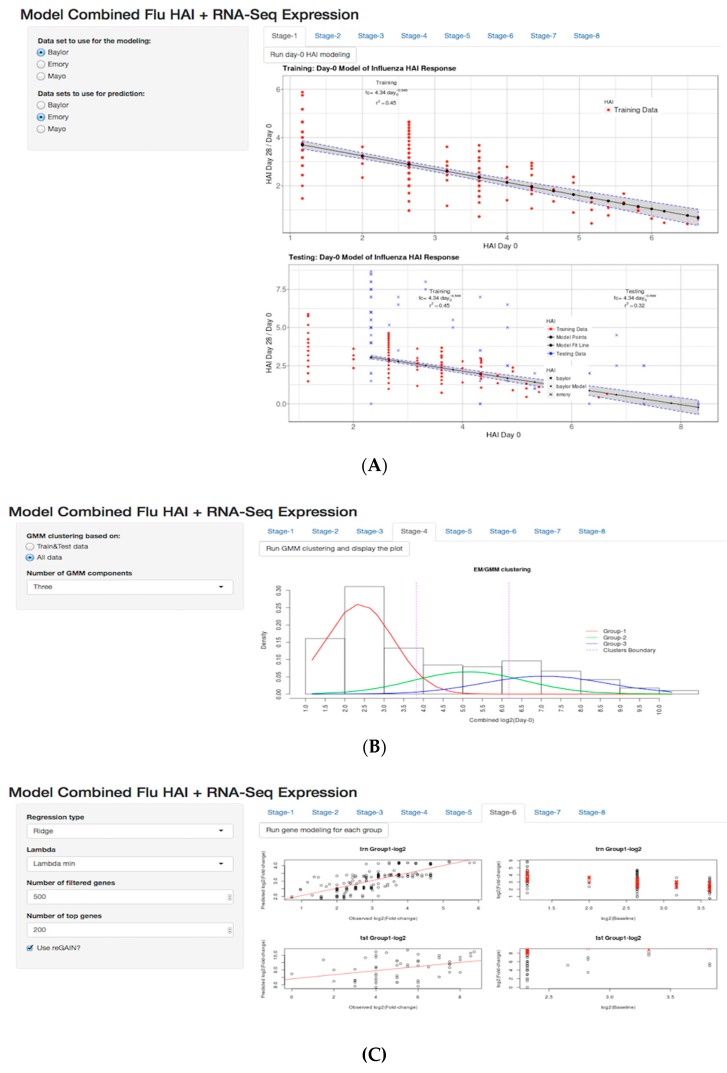
R Shiny App for modeling of flu vaccine HAI response. Sidebar panel displays the settings and main panel displays the result of modeling. (**A**) Sidebar panel displays options specifying the training and testing datasets. (**B**) Sidebar panel displays customization of the number of clusters, (**C**) selection of the regression modeling method, and (**D**) the number of genes filtered, including a download button for top 200 genes, where users are able to select the train and test data sets, The user can run the analysis for each tab and produce plots. Pre−loaded datasets are available by default in RData format. The web based app and source code are available at https://github.com/insilico/predictHAI and http:// insilico.utulsa.edu/predictHAI.

**Table 1 microorganisms-07-00079-t001:** Influenza vaccine data used for training and validation. Demographic summary and number of subjects with available data.

GEO Acc#	Location	Male:Female	Age	HAI at Day 0 and 28	Gene Expression Array Data				
					Day 0	Day 1	Day 3	Day 7	Day 14
**GSE48018**	Baylor Male	111:0	19–41	111	111	110	101	x	109
**GSE48023**	Baylor Female	0:107	19–41	107	107	107	105	x	98
**SDY67**	Mayo	57:92	50–74	149	105	x	105	x	105
**GSE29619**	Emory 2007–2009	27:38	22–40	63	63	x	63	63	x
**GSE74817**	Emory 2009–2011	35:51	21–85	80	58	58	58	58	58

x—data was not available on the given day post-vaccination.

**Table 2 microorganisms-07-00079-t002:** Top 10 enriched Reactome pathways using the top 200 genes ranked by reGAIN interaction pairs from the Baylor gene expression discovery data. The first column indicates the name of each pathway. The second column provides a description of the biological activity of the pathway. The third column provides the number of genes overlapping between the pathway and query list. The fourth and fifth columns show the p-value using a hypergeometric test and False Discovery Rate (FDR) of the enrichment, respectively.

Gene Set Name	Description	Overlap Genes	*p*-Value	FDR
Immune system	Genes involved in immune system	25	5.6 × 10^−13^	3.78 × 10^−10^
Adaptive immune system	Genes involved in adaptive immune system	15	1.81 × 10^−8^	6.1 × 10−^5^
Class I MHC mediated antigen processing presentation	Genes involved class I MHC mediated antigen processing and presentation	10	1.79 × 10^−7^	4.02 × 10^−5^
Antigen processing ubiquitination proteasome degradation	Genes involved in antigen processing: ubiquitination and proteasome degradation	9	4.38 × 10^−7^	7.39 × 10^−5^
Metabolism of RNA	Genes involved in metabolism of RNA	10	2.14 × 10^−6^	2.89 × 10^−4^
Generic transcription pathway	Genes involved in generic transcription pathway	10	3.8 × 10^−6^	4.27 × 10^−4^
Signaling by the B cell receptor BCR	Genes involved in signaling by the B cell receptor (BCR)	6	2.01 × 10^−5^	1.94 × 10^−3^
Mitotic G1_G1/S phases	Genes involved in mitotic G1-G1/S phases	6	3.23 × 10^−5^	2.34 × 10^−3^
Innate immune system	Genes involved in innate immune system	8	3.37 × 10^−5^	2.34 × 10^−3^
Regulation of mRNA stability by proteins that bind AU rich elements	Genes in involved in regulation of mRNA stability by proteins that bind AU-rich elements	5	3.47 × 10^−5^	2.34 × 10^−3^

**Table 3 microorganisms-07-00079-t003:** Table of overlap genes from our analysis in the first seven enriched pathways from Table 2.

Immune System	Adaptive Immune System	Class I MHC Mediated Antigen Processing Presentation	Antigen Processing Ubiquitination Proteasome Degradation	Metabolism of RNA	Generic Transcription Pathway	Signaling by the B Cell Receptor BCR
BCL2	CD36	CD36	DET1	CNOT10	MAML2	ORAI1
CASP1	CRTAM	DET1	HUWE1	DDX20	MED31	PIK3R1
CD36	CTSF	HUWE1	LRSAM1	EXOSC2	RBL1	PSMA4
CRTAM	DET1	LRSAM1	PJA1	EXOSC4	RORA	PSMC5
CTSF	HUWE1	PJA1	PSMA4	PSMA4	ZNF160	PSMF1
DET1	LRSAM1	PSMA4	PSMC5	PSMC5	ZNF180	SOS1
FLNB	ORAI1	PSMC5	PSMF1	PSMF1	ZNF197	
HUWE1	PIK3R1	PSMF1	RNF4	RBM8A	ZNF430	
IL6R	PJA1	RNF4	UBOX5	RPLP1	ZNF517	
LRSAM1	PSMA4	UBOX5		SNRPD3	ZNF589	
MAP2K7	PSMC5					
MAPK1	PSMF1					
MAPK13	RNF4					
NLRP3	SOS1					
ORAI1	UBOX5					
PIK3R1						
PJA1						
PSMA4						
PSMC5						
PSMF1						
RNF4						
RPS6KA3						
SOS1						
TLR1						
UBOX5

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
