# Peer review of "Multi-Level Model to Predict Antibody Response to Influenza Vaccine Using Gene Expression Interaction Network Feature Selection"

_microorganisms, 2019, doi:10.3390/microorganisms7030079_

Reviewer 1 Report

In the current manuscript the authors describe a refined methodology for the prediction of the IAV vaccine outcome base on pre-vaccination HAI titters and gene interaction networks. The manuscript is dense and not easily accessible if ones is not familiar with the described methodologies. Substantial changes must be made to facilitate the reading to non-experts in the field that are, nonetheless, interested in IAV vaccination and antibody response, especially considering the wide scope of the the journal.

These changes must include:

- A better description of the objectives and results in the abstract section.

- An extended description of the methodology (not necessarily on the Mat and Meth section), the journal does not have length limits  please take advantage of it.

- A better description of the databases used (please include a brief description of how the samples were collected and processed).

-Please describe the nature of the gene expression dated fed into the pipeline.

On figure 2. On the bottom left panel should read Testing: Mayo Data instead of Training: Mayo data.

Figure 4. the figure legend should include the name of each group and not just group 1, 2 and 3.

The tool described in Appendix A would be much more useful if input from other sources could be included and a prediction (of a vaccine performance) obtained. If these changes are made include the description of the tool in the main body of the manuscript and not as an appendix.

p { margin-bottom: 0.1in; line-height: 115%; }

Author Response

In the current manuscript the authors describe a refined methodology for the prediction of the IAV vaccine outcome base on pre-vaccination HAI titters and gene interaction networks. The manuscript is dense and not easily accessible if ones is not familiar with the described methodologies. Substantial changes must be made to facilitate the reading to non-experts in the field that are, nonetheless, interested in IAV vaccination and antibody response, especially considering the wide scope of the journal.

Authors’ response:

We thank the reviewer for these constructive comments to improve the manuscript. Below is our response to each specific comment. Changes in the manuscript are colored blue.

- A better description of the objectives and results in the abstract section.

Based on this comment, we improved the abstract by noting that the GAIN feature selection approach improved model generalizability while identifying genes enriched for immunologically relevant pathways. We note that using a multi-level approach, starting with a baseline HAI model and stratifying based on baseline HAI, allows for more targeted gene-based modeling.

- An extended description of the methodology (not necessarily on the Mat and Meth section), the journal does not have length limits  please take advantage of it.

In the introduction and methods, we extend the description and better explain the methodology for a broader audience.

- A better description of the databases used (please include a brief description of how the samples were collected and processed). Please describe the nature of the gene expression data fed into the pipeline.

We provide additional detail on the source and nature of all data sets used in the analysis.

On figure 2. On the bottom left panel should read Testing: Mayo Data instead of Training: Mayo data.

Thank you for noticing this detail. That is fixed.

Figure 4. the figure legend should include the name of each group and not just group 1, 2 and 3.

The tool described in Appendix A would be much more useful if input from other sources could be included and a prediction (of a vaccine performance) obtained. If these changes are made include the description of the tool in the main body of the manuscript and not as an appendix.

Thank you for pointing this out. We have added a more descriptive legend to Fig. 4 that explains that the groups are lower, intermediate and high baseline titer groups.

We agree the ability to upload other sources of data would increase the flexibility and utility of the tool, and we are working on getting resources (a dedicated server with sufficient capacity) to host the tool in such a way that would allow users to upload additional data and perform the machine learning analysis. In principle the tool has this capability and we have shared the source code on github for others to implement. We have added a description of the tool to the main body of the manuscript. 

Reviewer 2 Report

This manuscript focuses on the the important problem of understanding antibody responses to influenza vaccination.  In this work, machine learning methods were applied predict flu vaccine response based on gene expression levels and pre-vaccine antibody titers. Strengths of the work include access to multiple large cohorts of samples and associated pre-vaccination measures of HAI.  Despite these strengths, the overall approach and results are difficult to appreciate because of presentation and writing style.  A focus on details of machine learning approaches and terminology overshadows the results and potential relevance to the vaccine community. It was also unclear what was gained from the study; the key result that B cell receptor signaling and immune relevant pathways were enriched in vaccine samples does not seem surprising.  Perhaps a different presentation style would better clarify value and biological/medical relevance. 

Author Response

This manuscript focuses on the important problem of understanding antibody responses to influenza vaccination. In this work, machine learning methods were applied predict flu vaccine response based on gene expression levels and pre-vaccine antibody titers. Strengths of the work include access to multiple large cohorts of samples and associated pre-vaccination measures of HAI. Despite these strengths, the overall approach and results are difficult to appreciate because of presentation and writing style. A focus on details of machine learning approaches and terminology overshadows the results and potential relevance to the vaccine community. It was also unclear what was gained from the study; the key result that B cell receptor signaling and immune relevant pathways were enriched in vaccine samples does not seem surprising. Perhaps a different presentation style would better clarify value and biological/medical relevance.

Authors’ response:

We thank the reviewer for these constructive comments. We revised the background with attention to clarifying the relevance to the vaccine community, and we revised the results and discussion to clarify the biological value of the approach. We find evidence that the gene interaction feature selection approach improves model generalizability by looking at ratios of interacting genes while identifying genes enriched for immunologically relevant pathways. Another advantage is that by starting with a baseline HAI model and stratifying based on baseline HAI, the multi-level approach is able to use more targeted gene-based modeling.

In the previous draft, we focused on discussing the BCR pathway because it was significant in another multi-cohort machine learning analysis of flu vaccine immune response. This suggests a common underlying gene module is being detected by both approaches. However, other interesting pathways were found by our approach, which we discuss in the revised manuscript, along with other points of relevance.

Reviewer 3 Report

This manuscript described a multi-layer machine learning modeling to predict the influenza vaccine response based on pre-vaccination antibody titers and gene expression data. The theoretical foundation of this modeling is fine.However, the presentation of the methodology and the delivery of results were not clearly written in this paper. Thus, it is required for authors to conduct a major revision of the manuscript.

Main points needed to address:

Please provide the background, summary and rational for machine learning modeling and vaccine response prediction.

It was confusing where line 52-54 mentioned to keep only "high" and "low" responders, but then in the overall approach section (line 85-87) mentioned to keep only "low" and "median" for reGAIN method.

When presenting results, the evaluation of test data set were not given in details. Please add.

When presenting result, the evaluation and comparison of non-reGAIN feature selection and reGAIN feather selection method were not provided. 

The discussion is poor. Please provide more discussion on biological meaning of the results, limitations and advantages of current machine learning approach.

Author Response

This manuscript described a multi-layer machine learning modeling to predict the influenza vaccine response based on pre-vaccination antibody titers and gene expression data. The theoretical foundation of this modeling is fine. However, the presentation of the methodology and the delivery of results were not clearly written in this paper. Thus, it is required for authors to conduct a major revision of the manuscript.

Main points needed to address:

Please provide the background, summary and rationale for machine learning modeling and vaccine response prediction.

Response: We have added rationale and justification for the proposed machine learning model strategy to the introduction. Changes in the manuscript are colored blue.

It was confusing where line 52-54 mentioned to keep only "high" and "low" responders, but then in the overall approach section (line 85-87) mentioned to keep only "low" and "median" for reGAIN method.

Response: We agree this could be confusing. In the introduction (52-54), we were describing the adjusted MFC method used in a previous study to place our method in the context of previous literature. However, our approach – rather than discretize the response – attempts to use all variation in HAI response. We discretize the baseline (pre-vaccine) HAI values and we group subjects into low, medium and high prior exposure groups because we expect different regulatory mechanisms to be at play for these groups. For the high prior-exposure group, we do not train a gene-based model because most of the variation in the HAI response is already explained by a simple day-0 HAI model.    

When presenting results, the evaluation of test data set were not given in details. Please add. When presenting result, the evaluation and comparison of non-reGAIN feature selection and reGAIN feather selection method were not provided. 

Response: We include a comparison of non-reGAIN feature selection in Fig. 6. For comparison we use coefficient of variation filtering. We find the reGAIN feature selection model through nested cross-validation leads to more variation explained (higher R2) in the independent data sets, suggesting that reGAIN helps find more biologically relevant combinations of genes with better generalizability.

The discussion is poor. Please provide more discussion on biological meaning of the results, limitations and advantages of current machine learning approach.

Response: We improve the discussion and add to our previous comments of the BCR pathway. This pathway was significant in another multi-cohort machine learning analysis of flu vaccine immune response, suggesting a common underlying gene module is detected by both approaches. However, other interesting pathways and results were found by our approach, which we discuss in the revised manuscript.

We also include limitations and advantages of the approach in the discussion. We find evidence that the gene interaction feature selection approach improves model generalizability by looking at ratios of interacting genes while identifying genes enriched for immunologically relevant pathways. Another advantage of the multi-level approach, starting with a baseline HAI model and stratifying based on baseline HAI allows for more targeted gene-based modeling. Additional variation may be explained by other types of feature selection and by boosting the machine learning models on individuals that lie farther away from the baseline HAI model prediction.

Round  2

Reviewer 1 Report

Accept in present form

Reviewer 2 Report

The overall clarity, presentation of the results, and discussion has improved greatly.  

Reviewer 3 Report

The revised version has adequately addressed my questions per first review. I would recommend to accept this paper for publish.